# Phospholipase A$_2$ activity is required for immune defense of European (*Apis mellifera*) and Asian (*Apis cerana*) honeybees against American foulbrood pathogen, *Paenibacillus larvae*

**Gahyeon Jin, Md Tafim Hossain Hrithik, Eeshita Mandal, Eui-Joon Kil, Chuleui Jung, Yonggyun Kim** *

Department of Plant Medicals, Andong National University, Andong, Korea

* hosanna@anu.ac.kr

**Data Availability Statement:** All relevant data are within the paper and its Supporting Information files.

## Abstract

Honeybees require an efficient immune system to defend against microbial pathogens. The American foulbrood pathogen, *Paenibacillus larvae*, is lethal to honeybees and one of the main causes of colony collapse. This study investigated the immune responses of *Apis mellifera* and *Apis cerana* honeybees against the bacterial pathogen *P. larvae*. Both species of honeybee larvae exhibited significant mortality even at $10^2$ $10^3$ cfu/mL of *P. larvae* by diet-feeding, although *A. mellifera* appeared to be more tolerant to the bacterial pathogen than *A. cerana*. Upon bacterial infection, the two honeybee species expressed both cellular and humoral immune responses. Hemocytes of both species exhibited characteristic spreading behaviors, accompanied by cytoskeletal extension along with F-actin growth, and formed nodules. Larvae of both species also expressed an antimicrobial peptide called apolipophorin III (*ApoLpIII*) in response to bacterial infection. However, these immune responses were significantly suppressed by a specific inhibitor to phospholipase A$_2$ (PLA$_2$). Each honeybee genome encodes four PLA$_2$ genes (*PLA$_2$A ~ PLA$_2$D*), representing four orthologous combinations between the two species. In response to *P. larvae* infection, both species significantly up-regulated PLA$_2$ enzyme activities and the expression of all four PLA$_2$ genes. To determine the roles of the four PLA$_2$s in the immune responses, RNA interference (RNAi) was performed by injecting gene-specific double stranded RNAs (dsRNAs). All four RNAi treatments significantly suppressed the immune responses, and specific inhibition of the two secretory PLA$_2$s (*PLA$_2$A* and *PLA$_2$B*) potently suppressed nodule formation and *ApoLpIII* expression. These results demonstrate the cellular and humoral immune responses of *A. mellifera* and *A. cerana* against *P. larvae*. This study suggests that eicosanoids play a crucial role in mediating common immune responses in two closely related honeybees.

**Funding:** National Research Foundation (NRF) (2022R1A2B5B03001792) to Yonggyun Kim National Research Foundation (NRF) (2018R1A6A1A03024862) to Chuleui Jung.

**Competing interests:** The authors have declared that no competing interests exist.

## 1. Introduction

Insect pollinators sustain plant species and increase crop yields [1–3]. Indeed, a third of all crops are pollinator-dependent, and in particular are dependent on bees such as the honeybees [4]. For example, the economic value of pollination by honeybees was estimated to be over 18-fold greater than that of their honey production in Korea [5]. Honeybees are social insects that build nests in which different generations live together with divided labors among the workers, drones, and the queen, and in which immatures are five instar larvae and pupae [6]. The adult lifespan varies from a few weeks to several months or years depending on the caste differentiation and even season. The Asian honeybee, *Apis cerana*, is also an important pollinator that provides essential pollination services to agricultural plant communities in Asian countries [7]. However, in recent decades, the *A. cerana* populations have declined in many countries and subsequent reductions in pollination services greatly affect plant communities [8,9]. Among factors leading to declines in bee populations, the use of pesticides is considered to be the main cause [10]. However, in addition to the direct toxic actions of insecticides, sublethal doses have significant adverse effects on honeybee immunity [11].

American foulbrood is the most destructive bacterial disease of the honeybee larvae, in which only ten spores of the bacteria are sufficient to trigger a lethal infection [12,13]. Infection typically occurs through feeding to the larvae by worker bees and roughly 12 h after ingestion, the spores germinate in the larval gut epithelium to proliferate and kill the infected larva [14]. The dead larvae are then decomposed to form scales containing millions of the bacterial spores, which are spread around the hive by worker bees, leading to massive colony collapse [15].

Similar to other social insects, the honeybees defend against pathogenic microbes or eukaryotic parasites with communal defenses and individual immunity [16]. Communal defense represents hygienic behaviors like grooming and hive fever exhibited by young workers [17–19]. Individual immunity includes cellular and humoral responses in combination with a frontier barrier like a cuticle [16,20]. Suppression of these immune responses can lead to colony collapse [21]. For example, an exposure of queens to neonicotinoid pesticides reduced their total hemocyte number and impaired wound healing and antimicrobial peptide (AMP) production, leading to an immunosuppressed state [22,23]. The immunosuppression induced by insecticides then increases the susceptibility to the pathogens as was demonstrated in the larvae of *A. mellifera* exhibiting high mortality to *P. larvae*, after exposure to insecticides [24].

Insect immunity is innate and its recognition against pathogens is programmed in the genome by specific pattern recognition receptors [25]. Recent discovery of the infection-induced damage signal via dorsal switch protein 1 (DSP1) is additional to the innate insect recognition system [26,27]. The recognition signal is propagated to nearby effector tissues such as hemocytes and fat bodies [28]. A number of immune mediators have been identified and these include nitric oxide, cytokines, biogenic monoamines, and eicosanoids. Each plays a crucial role in activating immune effectors for various immune responses in insects [29]. Cross-talk between immune mediators occurs upon various pathogen infections, and eicosanoids play a central role in mediating immune signals with their chemical diversity [30]. These eicosanoids are likely to mediate the immune responses of the honeybees because nodule formation in response to bacterial infection was shown to be dependent on phospholipase $A_2$ ($PLA_2$) activity in *A. mellifera* [31].

$PLA_2$ catalyzes the committed step for eicosanoid biosynthesis and releases arachidonic acid (AA) from phospholipids [32]. Since the first $PLA_2$ was isolated from snake venom, a number of venomous or non-venomous $PLA_2$s have been identified and classified into at least

16 Groups (I-XVI) [33]. They are also classified into five major types: secretory PLA$_2$s (sPLA$_2$s: Groups I-III, V, IX, X, XI, XII, XIII, XIV, and XV), calcium-dependent intracellular PLA$_2$ (cPLA$_2$: Group IV), calcium-independent intracellular PLA$_2$ (iPLA$_2$: Group VI), lipo-protein-associated PLA$_2$ (LpPLA$_2$: Groups VII and VIII), and adipose phospholipase A$_2$ (AdPLA$_2$: Group XVI) [34]. However, PLA$_2$s encoded in honeybee genomes have not been analyzed in terms of either their identities or functions.

This study identified the PLA$_2$s encoded in two honeybee species, *A. mellifera* and *A. cerana*. The physiological functions of the PLA$_2$s were assessed in mediating cellular and humoral immune responses in the two species. Using the full genomes, this study proposed an immune signal pathway associated with eicosanoid biosynthesis in honeybees.

## 2. Materials and methods

### 2.1. Honeybee larvae collection and diet preparation

The experiment was carried out on two species, *A. mellifera* and *A. cerana*, which were reared in the experimental apiary of Andong National University. The honeybees were raised in non-cultivated areas where exposure to agricultural chemicals such as pesticides including neonico-tinoids was minimized. The experimental larvae were collected from beehives at the experimental apiary. To isolate the queen and a few workers, a cage was used to separate them on a new frame for 24 h. Afterward, the queens were released, and the frame containing newly laid eggs was returned to the hive with a protective cage for an additional 70 h. Under these rearing conditions, larvae underwent five instars (L1-L5). Larval stage used 2 days-old L5 individuals for immunological assays. Bioassays against the bacterial pathogen used L3 larvae.

### 2.2. Bacterial culture

The bacterial pathogen, *P. larvae*, used in this study was obtained from Korean Agricultural Colony Collection (KACC, RDA, Wanju, Korea) with an accession number of NZ_CP019687.1. The bacterium was grown in brain heart infusion medium (BHI: Millipore, Burlington, MA, USA) for 18 h at 30°C with shaking at 180 rpm.

### 2.3. Chemicals

Arachidonic acid (AA, 5,8,11,14-eicosatetraenoic acid) and dexamethasone (DEX, (11β, 16α)-9-fluoro-11,17,21-trihydroxy-16-methylpregna-1,4-diene-3) were obtained from Sigma-Aldrich Korea (Seoul, Korea). They were dissolved in dimethyl sulfoxide (DMSO) to prepare test solutions. To prevent any melanization of hemolymph, an anticoagulant buffer (ACB) was prepared with 186 mM NaCl, 17 mM Na$_2$EDTA, and 41 mM citric acid and adjusted to pH 4.5 with acetic acid. Phosphate-buffered saline (PBS) was prepared with 100 mM phosphate plus 0.75% NaCl and pH adjusted to 7.4 with NaOH.

### 2.4. Hemocyte counts

Ninety μL of ACB was mixed with the hemolymph samples collected from 25 individuals of L5 instar larvae. The collected hemolymph suspension was then centrifuged at $1,000 \times g$ for 3 min to obtain the cell pellet which was then resuspended in 40 μL of TC-100 insect tissue culture medium (Hyclone, Daegu, Korea). Total hemocyte count (THC) and differential hemocyte count (DHC) were determined with a hemocytometer. The classification of hemocyte types was based on the morphological characteristics described by Lavine and Strand [35]. Each treatment was replicated independently three times.

## 2.5. Hemocyte-spreading assay

L5 instar larvae of both honeybee species were used for the hemocyte-spreading behavior. Hemocytes were collected using ACB as described above and incubated on ice for 20 min. The diluted hemolymph was then centrifuged at $1,000 \times g$ for 3 min at 4˚C to get the pellet, which was re-suspended in 300 μL of filter-sterilized TC-100 insect cell culture medium. Ten μL of hemocyte suspension was laid on a glass coverslip. After removing supernatant, hemocytes were fixed with 4% paraformaldehyde for 10 min at 25˚C. After washing three times with filter-sterilized PBS, hemocytes were then permeabilized with 0.2% Triton-X in PBS for 2 min at 25˚C. After washing three times, hemocytes were incubated with 5% skim milk for 10 min at 25˚C and subsequently with fluorescein isothiocyanate (FITC)-tagged phalloidin in PBS for 60 min. After washing three times, the cells were incubated with 4′,6-diamidino-2-phenylindole (DAPI, 1 mg/mL). Finally, after washing twice in PBS, the cells were observed under a fluorescence microscope (DM2500, Leica, Wetzlar, Germany) at $400 \times$ magnification. Hemocyte-spreading was determined based on the extension of F-actin growth beyond the original cell boundary. Scoring the spread cells was performed by counting the cells exhibiting F-actin growth among 100 randomly chosen cells. Each treatment was replicated three times with independent hemocyte preparations.

## 2.6. Nodulation assay

Hemocyte nodule formation was evaluated in L5 larvae of *A. mellifera* and *A. cerana*. Each larva was injected with 1 μL of overnight-cultured *P. larvae* ($5 \times 10^7$ cells/mL) and 1 μL of a test chemical (DEX or AA, 1 μg per larva), into the hemocoel through the proleg using a micro-syringe (Hamilton, Reno, NV, USA). For controls, DMSO was injected along with the bacteria. The injected larvae were then incubated at 25˚C for 8 h to reduce any variation in the nodule formation depending on varying ambient temperatures. After incubation, the larvae were dissected to count the melanized nodules under a microscope (Stemi SV11, Zeiss, Jena, Germany) at 50× magnification. In each treatment, nine larvae were assessed.

## 2.7. Bioinformatics to predict PLA$_2$ genes

Eight PLA$_2$ genes were retrieved with accession numbers XM_016916293.2 (*Am-PLA$_2$A*), XM_393116.7 (*Am-PLA$_2$B*), XM_624469.6 (*Am-PLA$_2$C*), JQ_900376.1 (*Am-PLA$_2$D*), XM_017064190.2 (*Ac-PLA$_2$A*), XM_017059900.2 (*Ac-PLA$_2$B*), XM_017051624.2 (*Ac-PLA$_2$C*), XM_ 017066061.2 (*Ac-PLA$_2$D*). MEGA10 was used to construct a phylogenetic tree using the Neighbor-joining method. Bootstrap values at each branch were calculated with 1,000 repeats. Interpro (http://www.ebi.ac.uk/interpro/) and Expasy (WWW.expasy.com) were used to predict domain and signal peptides.

## 2.8. RNA extraction, cDNA construction, and qPCR

L5 instar larvae, 2–3 day old pupae, and young worker bees less than 1 week old after emergence were used for total RNA extraction after removing their intestines to avoid any contamination derived from non-target organisms using Trizol reagent (Invitrogen, Carlsbad, CA, USA) according to the manufacturer's instructions. Extracted RNA was used for synthesizing complementary DNA (cDNA) using RT-premix (Intron Biotechnology, Seoul, Korea) containing an oligo-dT primer. Synthesized cDNA was quantified with a spectrophotometer (NanoDrop, Thermo Fisher Scientific, Wilmington, DE, USA). Synthesized cDNA (80 ng per μL) was used as a template for quantitative PCR (qPCR) using gene-specific primers (S1

Table). qPCR was performed using SYBR Green real-time PCR master mixture (Toyobo, Osaka, Japan) according to the guidelines of Bustin et al. [36] on a real-time PCR system (Step One Plus Real-Time PCR System, Applied Biosystems, Singapore). The reaction mixture (20 μL) contained 10 μL of Power SYBR Green PCR Master Mix, 1 μL of cDNA template (80 ng), and 1 μL each of forward and reverse primers, and 7 μL of deionized distilled water. The temperature program for qPCR began with 95°C heat treatment for 10 min followed by 40 cycles of denaturation at 94°C for 30 s, annealing at 52°C for 30 s, and extension at 72°C for 30 s. The ribosomal protein gene, *RL32*, was used as an endogenous control. Each treatment was replicated three times with independent samples. Expression analysis of qPCR was calculated by comparative CT method [37].

### 2.9. Virulence assay of *P. larvae* against the honeybee larvae

Test L3 larvae were fed with an artificial diet containing royal jelly (44.3%), glucose (5.3%), fructose (5.3%), yeast (0.9%), and sterilized water (44.3%) [38]. Diet was placed into wells of 48 well-plates containing individual larva. For the virulence test treatment, the artificial diet included *P. larvae* at the pre-determined bacterial concentrations in colony-forming unit (CFU). Ten larvae were used in each concentration treatment and replicated three times.

### 2.10. dsRNA preparation and RNAi treatment

T7 promoter sequence was linked to gene-specific primers at the 5' end. Using these primers, a partial PLA$_2$ gene was amplified. The PCR product was then used to generate double-stranded RNA (dsRNA) using the Megascript RNAi Kit (Ambion, Austin, TX, USA). The dsRNA was mixed with a transfection reagent (Metafectene Pro, Biontex, Planegg, Germany) in 1:1 ratio. Late L4 larvae were used for RNAi treatment. To administer the dsRNA, a microsyringe (Hamilton, Reno, NV, USA) was employed to inject 1 μg of dsRNA per larva. A green fluorescence protein (*GFP*) was used as a control to prepare dsRNA. Each treatment was replicated three times using independent RNA preparations.

### 2.11. Statistical analysis

All experiments in this study were conducted in three individual replications. The results were plotted using Sigma plot 10.0. Statistical analysis was performed using PROC GLM of the SAS program [39] with a one-way analysis of variance. Significant differences among the means were determined using the LSD test at a Type I error of 0.05, indicated by different letters.

## 3. Results

### 3.1. Comparative analysis of *P. larvae* virulence against honeybees

Bacterial administration of *P. larvae* to larvae used a feeding method, in which live bacterial cells were incorporated in the artificial diet. The fed larvae suffered from bacterial pathogenicity and some died with a blackened cadaver (Fig 1A). Virulence was dependent on the incubation time and *A. cerana* appeared to be more susceptible than *A. mellifera* in the median lethal time (LT$_{50}$): 3.65 days for *A. mellifera* and 2.48 days for *A. cerana* (Fig 1B) at the bacterial concentration ($10^5$ cfu/mL) by feeding. This differential susceptibility between two honey bee species was also appeared in the median lethal dose (LC$_{50}$): $1.1 \times 10^5$ cfu for *A. mellifera* and $2.4 \times 10^4$ cfu for *A. cerana* (Fig 1C) at 5 days after bacterial treatment. However, these two median values were not statistically different at Type error = 0.05.

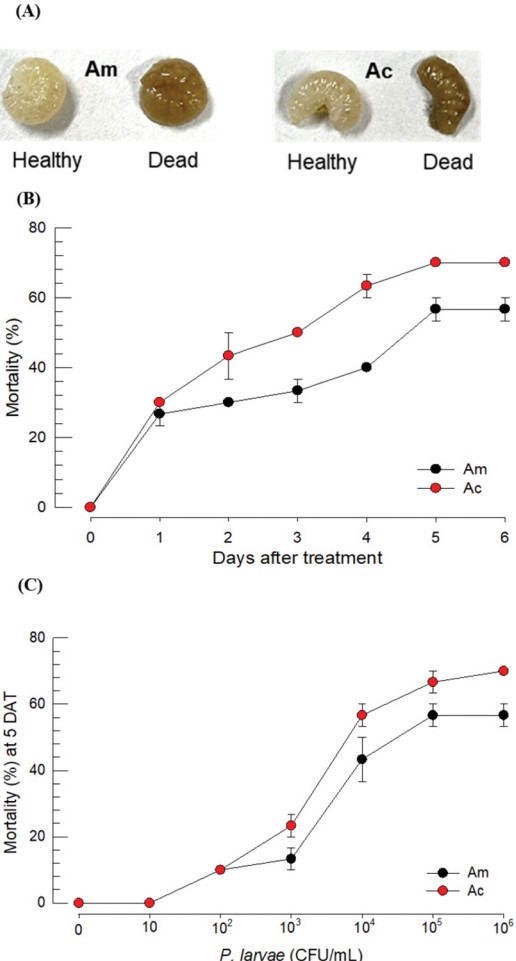

**Fig 1. Relative virulence of *P. larvae* against honeybee larvae of *A. mellifera* ('Am') and *A. cerana* ('Ac').** (A) Pathogenic symptom of the infected larvae. (B) Time-mortality curves of two honeybee species infected with *P. larvae*. L2 larvae were treated with bacteria ($10^5$ cfu/mL) by feeding. (C) Dose-mortality curves at 5 days after bacterial treatment ('DAT'). An experimental unit consisted of 10 larvae. Each dose was replicated three times. Different letters above standard deviation bars indicate significant differences among means at Type I error = 0.05.

### 3.2. Comparative analysis of immune responses of two honeybees

Hemocytes were classified on the basis of cell morphology (Fig 2A). Total hemocyte counts were not significantly different between the two honeybees (Fig 2B). At least three different types of hemocytes were discriminated, most of which were granulocytes (> 80%) in both species (Fig 2C). Granulocytes and plasmatocytes in particular, showed spreading on the plastic surface. Specifically, granulocytes were spread in all directions around the entire cell contour while plasmatocytes were spread unequally in specific directions. Oenocytoid cells were unspread and had a small nucleus compared to cytoplasm. The bacterial treatment significantly increased the hemocyte-spreading behavior in both honeybee species (Fig 2D).

Hemocoelic injection of the bacteria into larvae stimulated nodule formation. The nodulation was dependent on the incubation time and reached maximal levels 8 h after the bacterial infection (Fig 3A). Most nodules were detected near the trachea and on the fat body (see inset photos). The kinetics of nodule formation was not much different in the two species in terms of the time to form the maximal number of nodules and the number of nodules. Moreover,

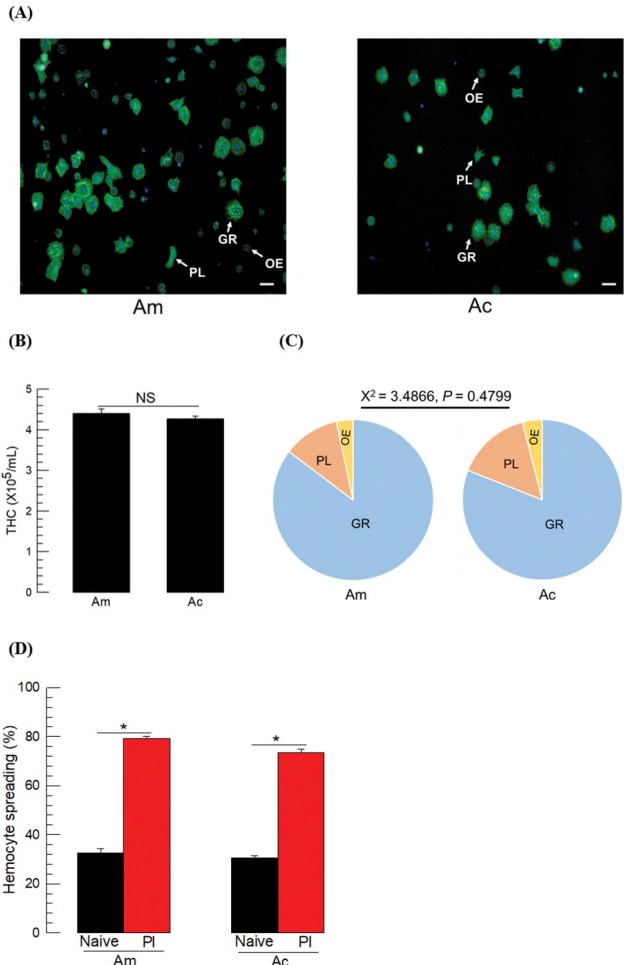

**Fig 2. Hemocytes and their behavior (*A. mellifera* ('Am') and *A. cerana* ('Ac')).** (A) Hemocyte types: Granulocyte ('GR'), plasmatocyte ('PL'), and oenocytoid ('OE'). Cytoplasm was stained with FITC against F-actin while nucleus was stained with DAPI. Scale bar indicates 10 μm. (B) Comparison of the total hemocyte count (THC) between two species. 'NS' stands for no significance. (C) Differential hemocyte counts. The statistical analysis was performed by $X^2$ test to compare the hemocyte composition between two species. (D) Hemocyte-spreading behavior. Each larva was injected with 1 μL of *P. larvae* ('Pl', $5 \times 10^4$ cells). Each measurement for the spreading behavior used 100 randomly chosen hemocytes. Each treatment was replicated three times by individual sample preparation. Different letters above the standard deviation bars indicate significant difference among means at Type I error = 0.05.

there was no difference in the numbers of nodules between the injections of live and dead *P. larvae* (Fig 3B).

Two AMPs were assessed in their expression levels after the bacterial infection of *P. larvae* (Fig 3C). The bacterial infection significantly up-regulated the gene expression of *apolipophorin III* (*ApoLpIII*), but not that of *defensin* in both honeybee species.

## 3.3. Eicosanoids mediate both cellular and humoral immune responses in honeybees

Immune responses are mediated by eicosanoids in insects [29]. To test this hypothesis in honeybees, eicosanoid biosynthesis was inhibited by dexamethasone (DEX, a specific inhibitor of $PLA_2$). DEX treatment significantly suppressed the formation of nodules in response to *P. larvae* infection in both species (Fig 4A). However, arachidonic acid (AA, a catalytic product of

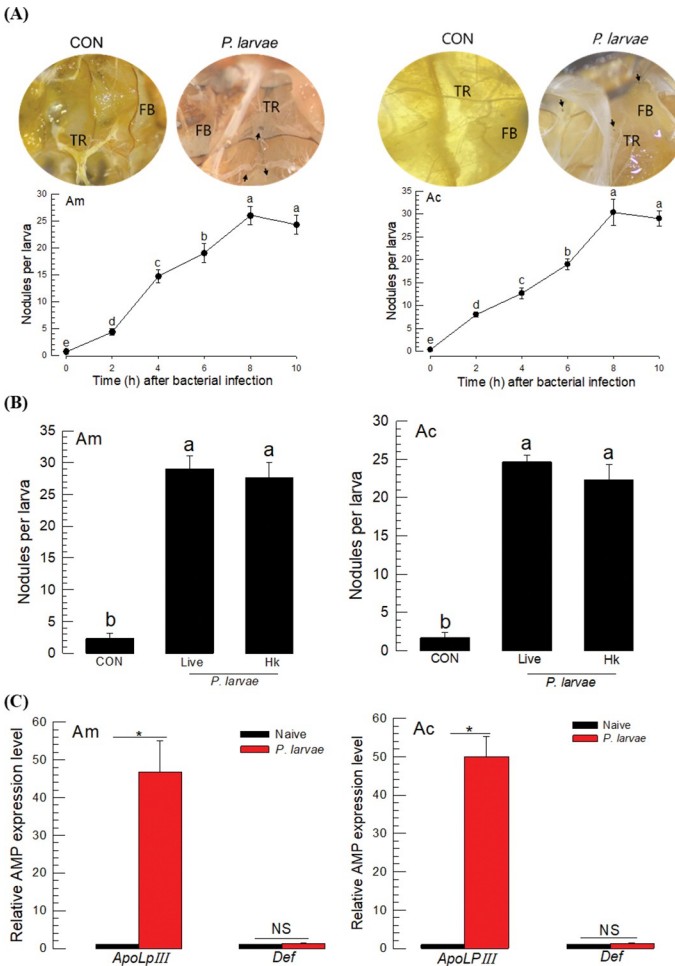

**Fig 3. Immune responses of honeybees *A. mellifera* ('Am') and *A. cerana* ('Ac').** (A) Cellular immune response observed by nodule formation upon infection of *P. larvae* by bacterial injection to L5 instar larvae at different time points. Each larva was injected with 1 μL of *P. larvae* ($5 \times 10^4$ cells). Each treatment was replicated three times. At 8 h after injection, the total number of nodules was counted. PBS was injected for controls. 'FB' and 'TR' stands for fat body and trachea, respectively. Arrows indicate nodules. (B) Comparison of live and heat-killed *P. larvae* in forming nodules in the honeybee larvae. Heat-killing ('Hk') treatment used 98°C for 20 min. Control ('CON') used sterilized PBS for injection. Different letters above the standard deviation bars indicate significant difference among means at Type I error = 0.05. (C) Humoral immune response assessed by expression of two AMP genes: *apolipophorin III* ('*ApoLpIII*') and defensin ('*Def*') at 8 h after bacterial injection. Each treatment was replicated three times. Asterisk stands for significant difference while 'NS' is no significant difference.

PLA₂) significantly rescued the inhibitory activity of DEX in both species. Similarly, *ApoLpIII* expression was also modulated by DEX and AA in both species (Fig 4B).

## 3.4. PLA₂ orthologs encoded in in two honeybee genomes

From each honeybee genome, four PLA₂ genes ($PLA_2A \sim PLA_2D$) were obtained and showed an orthologous relationship (Fig 5A). The four PLA₂s were classified into secretory (sPLA₂), calcium-independent (iPLA₂), and lysosomal (LPLA₂). PLA₂A and PLA₂B are secretory, while PLA₂C is calcium-independent and PLA₂D is lysosomal. PLA₂A and PLA₂B have signal peptides in their N termini and calcium-binding domains in addition to a catalytic domain (Fig 5B), which represent the typical domain composition of most sPLA₂s [40]. In contrast, PLA₂C has five ankyrin repeats in addition to a catalytic domain [41]. PLA₂D was classified into Group XV PLA₂s specific to lysosomal PLA₂s [42].

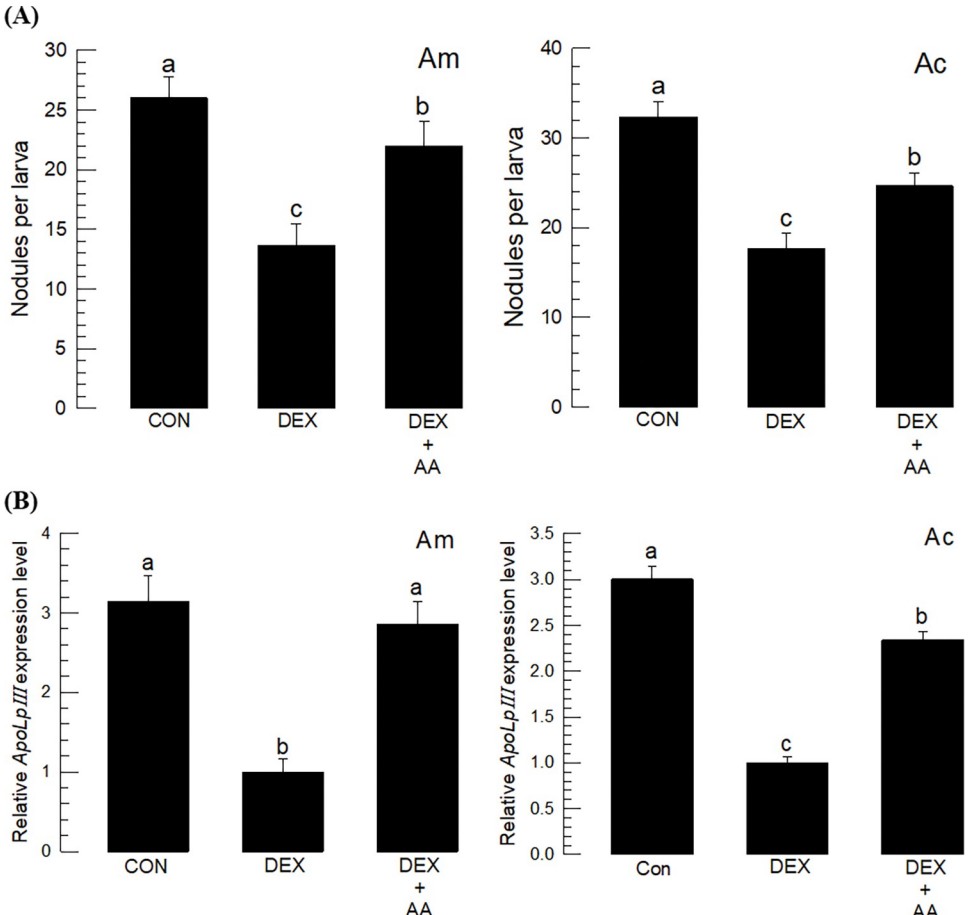

**Fig 4. Eicosanoid mediation of the immune responses in honeybee species: *A. mellifera* ('Am') and *A. cerana* ('Ac').** L5 instar larvae were immune-challenged by injection with 1 μL of *P. larvae* ($5 \times 10^4$ cells). (A) Inhibitory effect of dexamethasone ('DEX', 1 μg/larva) on nodule formation in response to bacterial infection. Rescue by addition of arachidonic acid (1 μg/larva). (B) Influence of DEX on *ApoLpIII* gene expression upon bacterial infection. Different letters above the standard deviation bars indicate significant differences among means at Type I error = 0.05.

### 3.5. Expression profile of four PLA$_2$ genes

Fig 6 shows that the four PLA$_2$ genes were expressed in different developmental stages of both honeybee species. Of note, *PLA$_2$D* was highly expressed in all developmental stages. In addition, PLA$_2$C was also relatively highly expressed like *PLA$_2$D* in both species. In contrast, the Group III *PLA$_2$A* and Group XII *PLA$_2$B* were expressed at low levels in all developmental stages in both species. However, *PLA$_2$B* was highly induced in adults.

### 3.6. Induction of PLA$_2$ expression upon *P. larvae* infection

Upon infection with *P. larvae*, PLA$_2$ activities were significantly up-regulated in larval and adult stages of the two honeybees (Fig 7A). The induction of elevated enzyme activity was further supported by up-regulation of PLA$_2$ gene expression in *A. mellifera* (Fig 7B) and *A. cerana*.

### 3.7. RNAi against PLA$_2$ genes increased the virulence of *P. larvae*

To clarify the role of immune-associated PLA$_2$(s) in honeybees, gene-specific dsRNAs were injected into larvae to suppress target PLA$_2$ genes. dsRNA effectively suppressed the

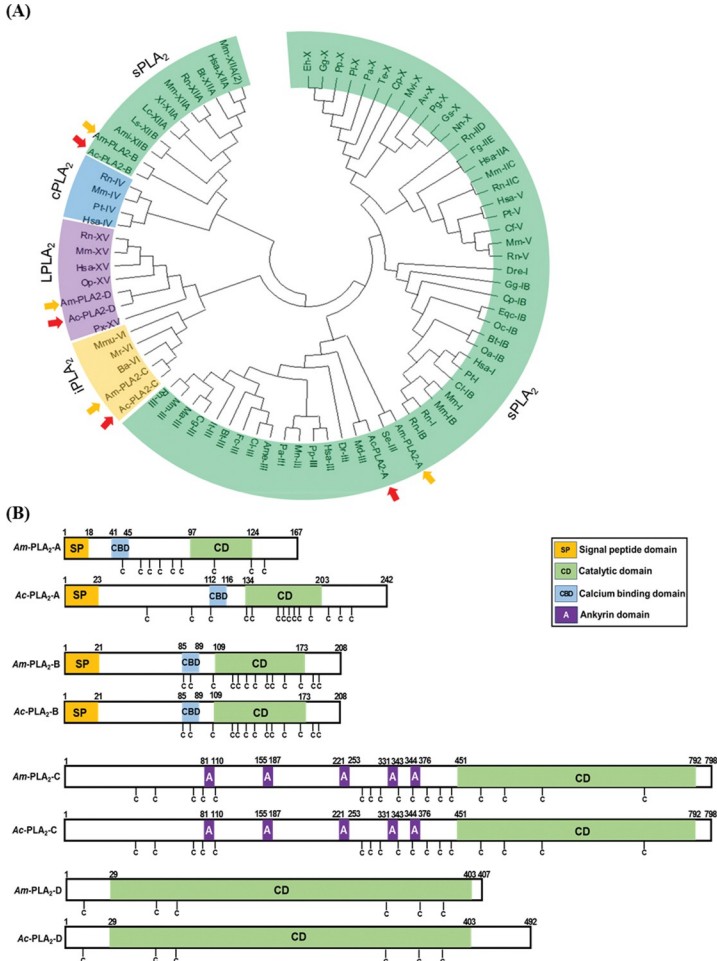

**Fig 5. Orthologs of four PLA$_2$ genes encoded in honeybee species: *A. mellifera* ('Am' in yellow arrows) and *A. cerana* ('Ac' in orange arrows).** (A) Phylogenetic tree of the honeybee PLA$_2$s with different PLA$_2$, groups delineated. Honeybee PLA$_2$s were clustered in four types: secretory (sPLA$_2$), lysosomal (LPLA$_2$), Ca$^{2+}$-independent cellular (iPLA$_2$), and Ca$^{2+}$-dependent cellular (cPLA$_2$). (B) Variation of functional domains among the honeybee PLA$_2$s. Cysteine ('C') residues are denoted.

expression levels of target genes in both species (S1 Fig). Under these conditions, the honeybee larvae were significantly suppressed in their ability to form nodules in response to immune challenge with *P. larvae* (Fig 8A). Although all four RNAi treatments were effective at suppressing the cellular immune response, RNAi treatments against *PLA$_2$A* or *PLA$_2$B* appeared to be more potent than those of *PLA$_2$C* or *PLA$_2$D*. Similarly, all four RNAi treatments were effective at suppressing induction of *ApoLpIII* in the face of bacterial challenge (Fig 8B). In this humoral immune response, RNAi treatments against *PLA$_2$A* or *PLA$_2$B* were much more potent at suppressing AMP gene expression than that of *PLA$_2$C* or *PLA$_2$D*.

## 4. Discussion

Immunosuppression makes honeybees highly susceptible to pathogens and can lead to colony collapse [43]. For example, the honeybees that are malnourished as a result of protein deficiency in their diet may be altered in specific components of the immune system, which can lead to fatal immunosuppression [44]. The American foulbrood disease caused by *P. larvae*

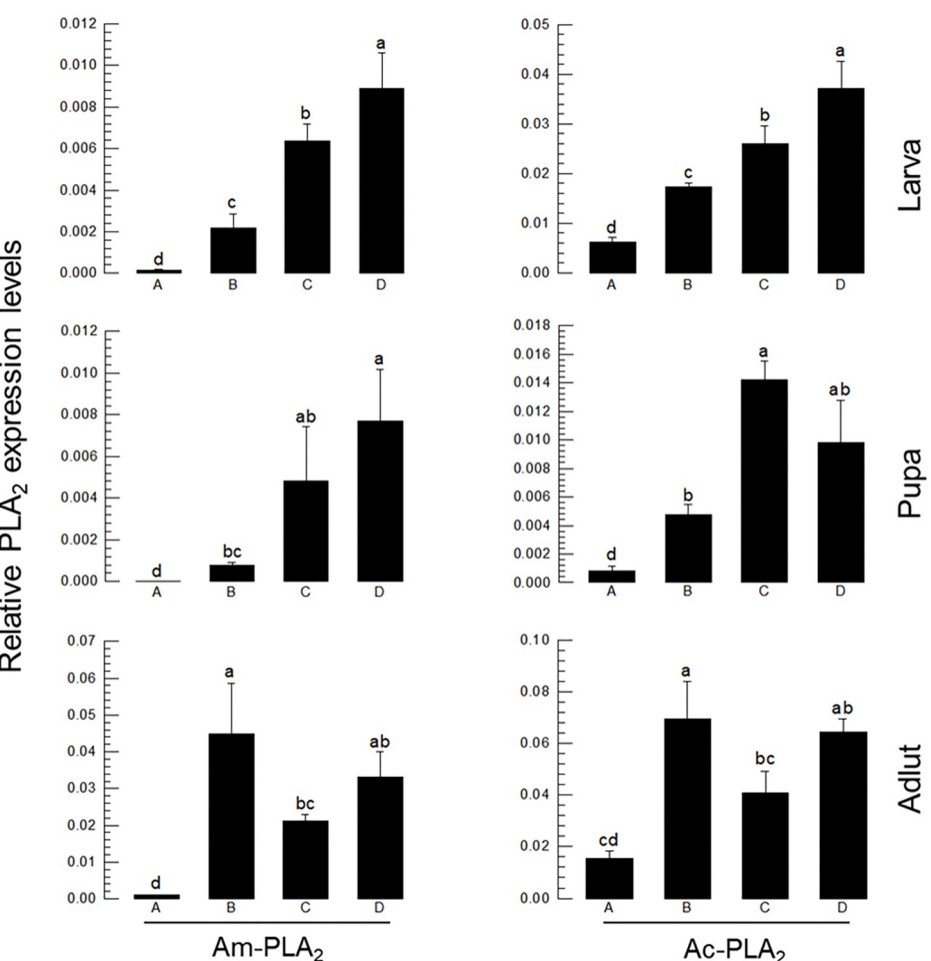

**Fig 6. Expression profile of four PLA$_2$s (PLA$_2$A, PLA$_2$B, PLA$_2$C, PLA$_2$D) in different developmental stages of honeybee species: *A. mellifera* ('Am') and *A. cerana* ('Ac').** Different letters above standard deviation bars indicate significant difference among means at Type I error = 0.05. Ribosomal protein *RPL32* was used as a reference gene. Each treatment was replicated three times with independent sample preparations.

becomes serious when honeybees are also suffering from immunosuppression after exposure to sublethal doses of insecticides [24]. This investigation has shown that eicosanoids play a crucial role in defending honeybees against American foulbrood disease.

Asian honeybees were shown to be more susceptible to the American foulbrood pathogen, *P. larvae* than the European honeybees. This supports the earlier comparative pathogenicity testing performed by Krongdang et al. [45]. However, bacterial virulence may be altered by the gut microbiota. In a similar Asian honeybee, *A. cerana japonica*, five species of gut bacteria exhibited a strong antagonistic activity against *P. larvae* growth and their relative abundance may modulate its bacterial virulence [46].

Both honeybee species exhibited similar hemocyte compositions in THC and DHC, in which granulocytes were predominant among the three different hemocytes that were identified. They also showed cellular immune responses upon challenge such as hemocyte-spreading behavior and nodule formation against *P. larvae* infection. In addition, they showed up-regulation of specific AMPs such as apolipophorin III (*ApoLpIII*). Collectively, these observations suggest that the two honeybee species share cellular and humoral immune responses. In

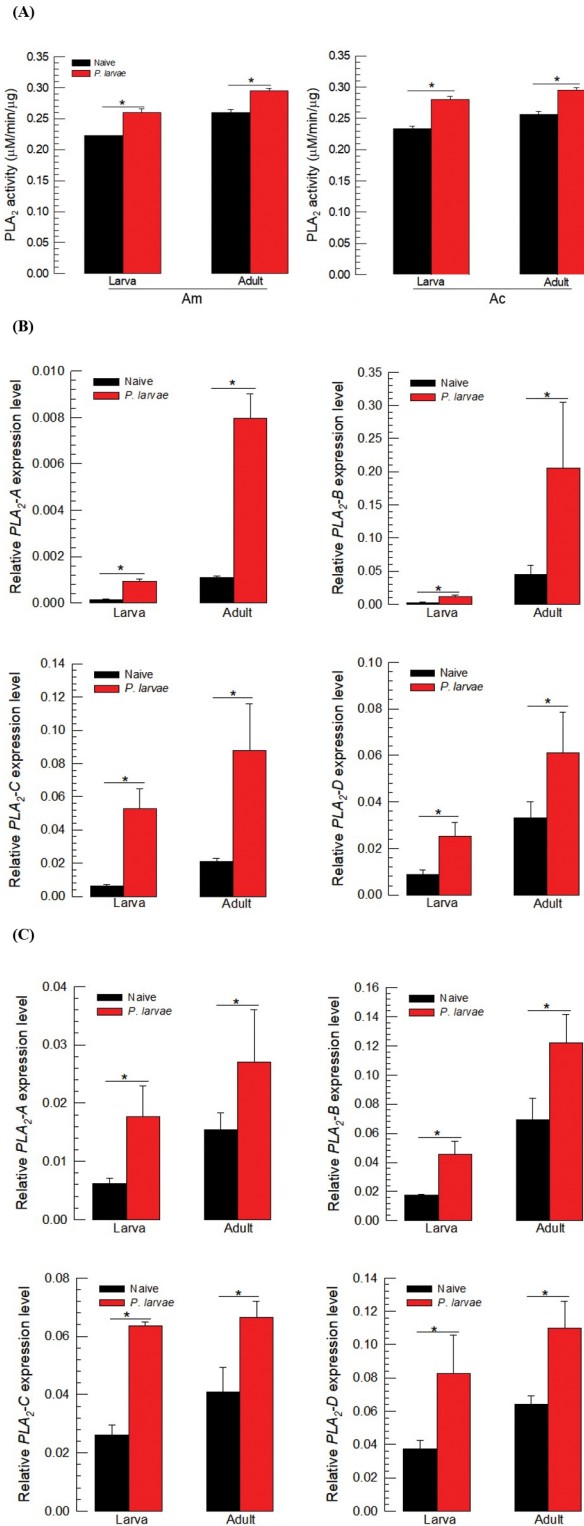

**Fig 7. Induction of PLA$_2$ enzyme activities and gene expression in response to immune-challenge by injection with 1 μL of *P. larvae* ($5 \times 10^4$ cells) in honeybee species: *A. mellifera* ('Am') and *A. cerana* ('Ac').** (A) Induction in PLA$_2$ enzyme activities in larvae and adults. The enzyme was extracted from the whole body of L5 instar larvae and the venom gland-removed adults. Different letters above standard deviation bars indicate significant difference among means at Type I error = 0.05. Induction of gene expression of four different PLA$_2$s after immune challenge in the larva and adult ('A') of Am (B) and Ac (C). The mRNA expression levels of each PLA$_2$ gene were measured at 8 h after

bacterial infection. Each treatment was replicated three times with independent sample preparations. Asterisks above standard deviation bars indicate significant difference among means at Type I error = 0.05. Ribosomal protein *RPL32* was used as a reference gene. Each treatment was replicated three times with independent sample preparations.

general, bees have an innate immune system supplemented by physical barriers (integument and peritrophic matrix), that include cellular, and humoral responses to defend against pathogens and parasites [47]. In particular, hemocytes represent the primary immune effectors for cellular immunity by exhibiting phagocytosis, nodule formation, and encapsulation, as well as the initiation of phenoloxidase (PO) that regulates coagulation or melanization [35]. Hemocytes of the honeybee are classified by their morphology with behavioral differences in lectin-binding and phagocytosis [48]. At least three types of hemocytes were observed here through all developmental stages and they included granulocytes, plasmatocytes, and oenocytoids [49]. However, the hemocyte composition varies with developmental stages and castes, and it is the phagocytic granulocytes that are most abundant at the larval stage but significantly diminish

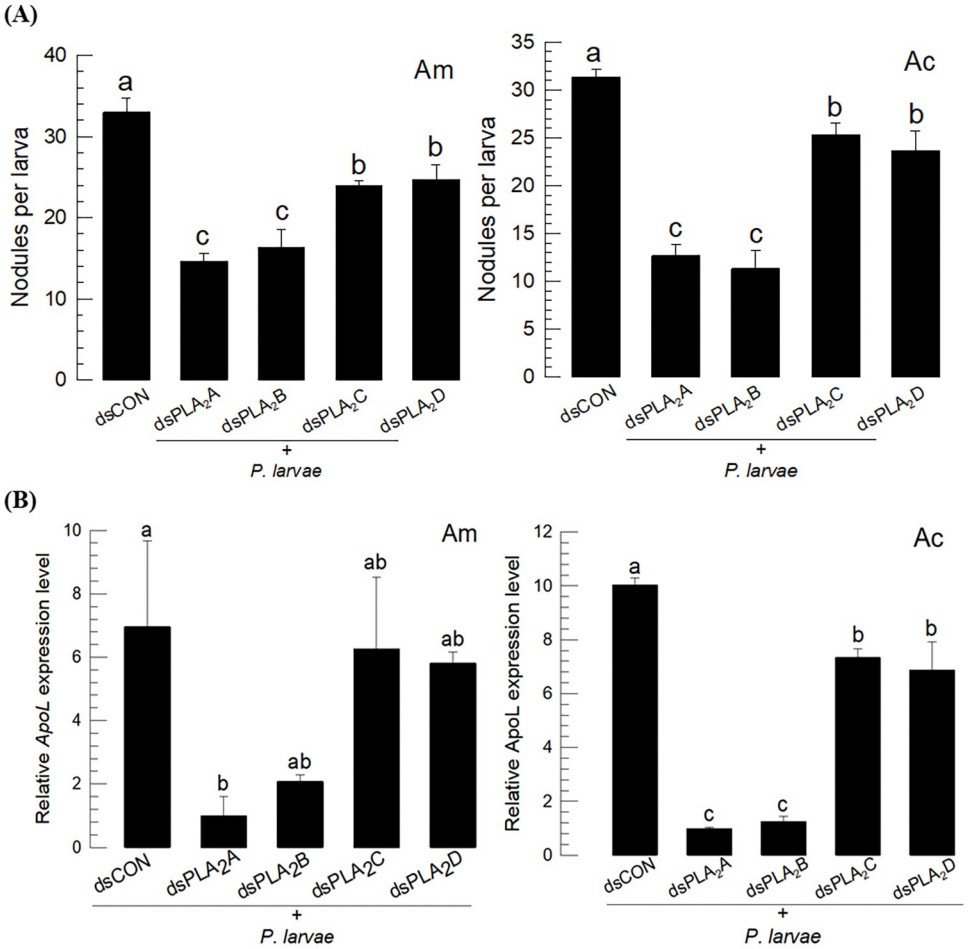

**Fig 8. Functional association of four honeybee PLA₂s with immune responses in *A. mellifera* ('Am') and *A. cerana* ('Ac').** Individual RNAi treatments were applied to specifically suppress *PLA₂-A* using dsPLA₂A, *PLA₂-B* using dsPLA₂B, *PLA₂-C* using dsPLA₂C, or *PLA₂-D* using dsPLA₂D. After 24 h post-injection of dsRNA, the immune challenge was performed by injecting 1 μL of *P. larvae* ($5 \times 10^4$ cells) into larvae or adults. Effects of the RNAi treatments on nodule formation (A) and *ApoLpIII* expression (B). *GFP* was used as a control dsRNA ('dsCON'). Each treatment was replicated three times. Ribosomal protein gene *RL32* was used as an internal control for RT-qPCR.

after adult emergence. Our current hemocyte analysis in both honeybee species supports the observation of Gabor et al. [49] at the larval stage of *A. mellifera*. In addition to the hemocytic activity, PO activity is required for nodulation. A serine protease called AccSp10 was identified in *A. cerana* and acted as a humoral factor to defend the microbial pathogens by suppressing bacterial growth and mediating melanization through activating PO activity [50]. In *A. mellifera*, 57 genes encode serine peptidases, however, only 6 show chymotrypsin-like specificity [51]. Thus, the two honeybee species studies here are genetically well programmed for expressing PO activity. Our AMP analysis indicated that *ApoLpIII* was highly up-regulated against *P. larvae*, suggesting a crucial role in defending against the bacterial pathogen. ApoLpIII was also demonstrated in *A. cerana* against other insect pathogens such as *Bacillus thuringiensis* and *Beauveria bassiana* [52]. This supports the immune-associated defense role of ApoLpIII against *P. larvae* infection in both honeybee species. In addition to the AMPs, vitellogenin (Vg, a yolk protein) plays a crucial role in defence against pathogens and neutralizing oxidative stress in honey bees [53]. *Vg* is expressed in fat body and venom glands of worker bees and Vg gene expression was highly up-regulated in response to pathogen infection and oxidative stress in *A. cerana* [54].

PLA$_2$ is associated with the cellular and humoral immune responses of honeybees. Dexamethasone treatment inhibited nodule formation and *ApoLpIII* expression while the addition of arachidonic acid (= a catalytic product of PLA$_2$) to the inhibitor treatment significantly rescued the immune responses. Eicosanoids are a subgroup of the oxygenated C20 polyunsaturated fatty acids and mediate many physiological processes in insects and other invertebrates [55]. They are usually synthesized from phospholipids by the catalytic activity of PLA$_2$ [56]. This suggests that the eicosanoids produced by PLA$_2$ catalysis promote the immune responses in honeybees.

Four PLA$_2$ genes are encoded by the two honeybee genomes. Of them, PLA$_2$A is known to be a honeybee venom component. PLA$_2$ and hyaluronidase enzymes account for 11~15% of the bee venom dry weight in *A. mellifera* [57,58]. Another sPLA$_2$ is PLA$_2$B, which is classified in Group XII. Similar Group XII PLA$_2$s were reported in other insects such as the bug, *Rhodnius prolixus* [59] and a moth, *Acrolepiopsis sapporensis* [60] and known to be associated with eicosanoid biosynthesis for reproduction. Most insects encode iPLA$_2$s, which are further divided into ankyrin-possessing or non-ankyrin iPLA$_2$s [29]. PLA$_2$C is classified into ankyrin-possessing iPLA$_2$. PLA$_2$D is classified as a lysosomal PLA$_2$ (LPLA$_2$) and is the first of this type known in insects. LPLA$_2$ is localized in the lysosome or late endosome in mammals [42] and its catalytic activity is optimal at acidic pH and is calcium-independent. Studies on the LPLA$_2$ null mouse suggest a role for the enzyme in the catabolism of pulmonary surfactant and it also has a role in host defense [61]. Our current study indicates that PLA$_2$D is the same in both honeybee species. PLA$_2$C and PLA$_2$D were highly expressed in larvae and pupae compared to the two other sPLA$_2$ genes in both species. This suggests that iPLA$_2$ and LPLA$_2$ of these honeybees may have important roles during immature development.

All PLA$_2$ genes were inducible in response to infection by *P. larvae* in both honeybees. Under RNAi inhibition of all four, the immune responses measured by nodulation and *ApoLpIII* expression were significantly suppressed. The immunosuppressive effects were more noticeable in the RNAi treatments specific to sPLA$_2$ genes compared to iPLA$_2$ or LPLA$_2$ genes. This strongly suggests that the two sPLA$_2$s are highly associated with immune responses in both honeybee species.

This study demonstrates the physiological role of PLA$_2$ in synthesizing eicosanoids, which in turn, mediate immune responses of the two honeybee species. This allows us to construct an eicosanoid signaling pathway in immune mediation from the honeybee genomes (Fig 9). Upon gut infection by virus, bacteria or fungi, a damage signal is triggered by the release of

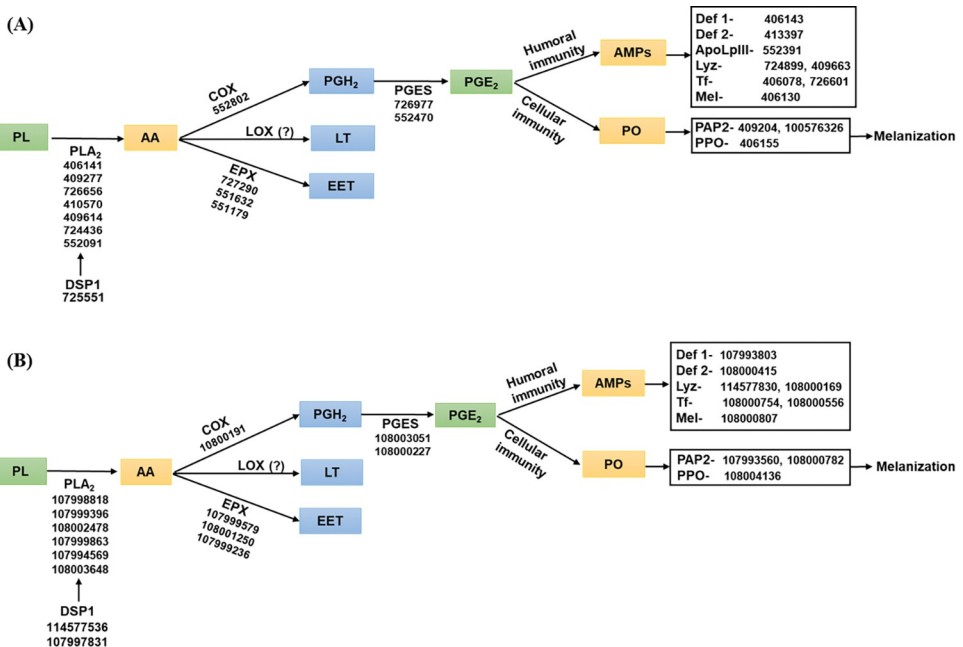

**Fig 9.** Immune responses mediated by eicosanoids in honeybee species, *A. mellifera* (A) and *A. cerana* (B). Upon immune challenge, a damage signal is triggered by dorsal switch protein 1 (DSP1). Release of DSP1 activates phospholipase $A_2$ ($PLA_2$) to initiate eicosanoid biosynthesis. The catalytic activity of $PLA_2$ produces arachidonic acid (AA), which is oxygenated by cyclooxygenase (COX) to produce prostaglandin $H_2$ ($PGH_2$). $PGH_2$ is then isomerized to $PGE_2$ by $PGE_2$ synthase (PGES). AA is alternatively oxygenated by lipoxygenase (LOX) to produce leukotriene (LT) or oxygenated by epoxygenase (EPX) to epoxyeicosatrienoic acid (EET). These eicosanoids including $PGE_2$ mediate cellular immune responses via phenoloxidase-activating protease (PAP) and phenoloxidase (PO), and humoral immune responses by antimicrobial peptides (AMPs) through defensin 1 (Def 1), defensin 2 (Def 2), apolipophorin III (ApoLpIII), lysozyme (Lyz), transferrin (Tf), and melittin (Mel). GenBank accessions are described in each signal components.

DSP1 from the midgut epithelium in insects [27,62,63]. DSP1 activates $PLA_2$ to catalyze AA release from phospholipids. AA is then used for synthesis of various eicosanoids through different oxygenases, with the resulting eicosanoids mediating cellular and humoral immune responses. Critically, all of the enzymes and effectors in the eicosanoid immune signaling pathway have now been shown to be encoded in the honeybee genomes.

## Supporting information

**S1 Fig. Suppression of $PLA_2$ gene expression by individual RNAi treatment.** (A) Change of $PLA_2$ expression levels in *A. mellifera* after injection (1 µg/larva) of dsRNA ('$dsPLA_2$-A', '$dsPLA_2$-B', '$dsPLA_2$-C', '$dsPLA_2$-D'). (B) Change of $PLA_2$ expression levels in *A. cerana* after injection (1 µg/larva) of dsRNA ('$dsPLA_2$-A', '$dsPLA_2$-B', '$dsPLA_2$-C', '$dsPLA_2$-D'). *GFP* was used as a control dsRNA ('dsCON').
(DOCX)

**S1 Table. List of primers used in this study.**
(DOCX)

## Author Contributions

**Conceptualization:** Yonggyun Kim.

**Data curation:** Gahyeon Jin, Md Tafim Hossain Hrithik, Eeshita Mandal, Eui-Joon Kil, Chuleui Jung.

**Formal analysis:** Gahyeon Jin, Md Tafim Hossain Hrithik, Eeshita Mandal, Eui-Joon Kil, Chuleui Jung.

**Funding acquisition:** Chuleui Jung, Yonggyun Kim.

**Investigation:** Gahyeon Jin, Md Tafim Hossain Hrithik, Eui-Joon Kil, Chuleui Jung, Yonggyun Kim.

**Methodology:** Gahyeon Jin, Md Tafim Hossain Hrithik, Eeshita Mandal, Eui-Joon Kil, Chuleui Jung, Yonggyun Kim.

**Project administration:** Yonggyun Kim.

**Resources:** Chuleui Jung, Yonggyun Kim.

**Software:** Gahyeon Jin, Md Tafim Hossain Hrithik, Eeshita Mandal, Eui-Joon Kil.

**Supervision:** Eui-Joon Kil, Chuleui Jung, Yonggyun Kim.

**Validation:** Gahyeon Jin, Md Tafim Hossain Hrithik, Eeshita Mandal, Yonggyun Kim.

**Visualization:** Gahyeon Jin, Md Tafim Hossain Hrithik.

**Writing – original draft:** Gahyeon Jin, Md Tafim Hossain Hrithik, Eeshita Mandal, Eui-Joon Kil, Yonggyun Kim.

**Writing – review & editing:** Chuleui Jung, Yonggyun Kim.

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
