## [Decision Letter · Decision Letter 0]

23 Oct 2023

PONE-D-23-26300Phospholipase A2 activity is required for immune defense of European (Apis mellifera) and Asian (Apis cerana) honeybees against the American foulbrood pathogen, Paenibacillus larvaePLOS ONE

Dear Dr. Kim,

Thank you for submitting your manuscript to PLOS ONE. After careful consideration by the editor and two reviewers, we feel that it has potential merit but does not meet PLOS ONE’s publication criteria as it currently stands. The reviewers raise a number of concerns that need to be addressed. Rigorous statistical analyses are key to the interpretation of all experiments and more detailed descriptions of the procedures are necessary. In many cases, a simple ANOVA is not appropriate to use because you have multiple factors, such as replicate and treatment to consider. For example, individual hemocytes within a replicate are not statistically independent. All statistical results have to be presented in the main text. Also, you need to substantiate your claim that dexamethasone is a specific inhibitor of PLA2 in honey bees, and that its effect can be rescued by arachidonic acid in honey bees. 

If you are prepared to address these concerns and revise your manuscript accordingly, please submit your revised manuscript by Dec 07 2023 11:59PM. If you will need more time than this to complete your revisions, please reply to this message or contact the journal office at plosone@plos.org. Please include the following items when submitting your revised manuscript:A rebuttal letter that responds to each point raised by the academic editor and reviewer(s). You should upload this letter as a separate file labeled 'Response to Reviewers'.A marked-up copy of your manuscript that highlights changes made to the original version. You should upload this as a separate file labeled 'Revised Manuscript with Track Changes'.An unmarked version of your revised paper without tracked changes. You should upload this as a separate file labeled 'Manuscript'.

We look forward to receiving your revised manuscript.

Kind regards,

Olav Rueppell

Academic Editor

PLOS ONE

4. We note that Figure 1A, 2A and 3A in your submission contain copyrighted images. All PLOS content is published under the Creative Commons Attribution License (CC BY 4.0), which means that the manuscript, images, and Supporting Information files will be freely available online, and any third party is permitted to access, download, copy, distribute, and use these materials in any way, even commercially, with proper attribution. For more information, see our copyright guidelines: http://journals.plos.org/plosone/s/licenses-and-copyright.

a. You may seek permission from the original copyright holder of Figure 1A, 2A and 3A to publish the content specifically under the CC BY 4.0 license. 

Reviewers' comments:

Reviewer's Responses to Questions

**Comments to the Author**

1. Is the manuscript technically sound, and do the data support the conclusions?

Reviewer #1: Partly

Reviewer #2: Partly

2. Has the statistical analysis been performed appropriately and rigorously? 

Reviewer #1: Yes

Reviewer #2: No

3. Have the authors made all data underlying the findings in their manuscript fully available?

Reviewer #1: Yes

Reviewer #2: Yes

4. Is the manuscript presented in an intelligible fashion and written in standard English?

Reviewer #1: Yes

Reviewer #2: No

5. Review Comments to the Author

Reviewer #1: This manuscript reports on the requirement of Phosphatase A2 (PLA2) activity for immune defense of Apis mellifera (European honeybee) and the Asian honeybee A. cerana, against the American foulbrood pathogen Paenibacillus larvae.

The authors analyzed the impact of infection by diet-feeding of different doses of the pathogenic bacteria P. larvae on the immune system of both species of honeybees A. cerana and A. mellifera. Different methods were applied to follow the cellular and systemic immune responses of these two species. The authors performed for example hemocyte counts, an hemocyte-spreading assay, nodulation formation, bioinformatic approaches to evidence the PLA2 gene diversity, and RNAi treatments. All results were performed in three individual replicates. The main conclusions of the study is reported that both species exhibited significant mortalities when infected by P. larvae by diet-feeding. Larvae of both species expressed Apolipophorin III (ApoLpIII) in response to the infection protocol used while the immune responses were significantly suppressed by a specific inhibitor of PLA2, PLA2 being observed up-regulated the expression of the 4 genes identified in both species. Finally, the authors observed that an approach of RNAi significantly suppressed (1) the immune responses and a specific inhibition of PLA2A and PLA2B genes (secretory forms of PLA2); and (2) nodule formation and ApoLpIII gene expression.

This study is of interest in the field of honeybee health, impact of pathogenic bacteria on both cellular and humoral immune responses in a comparative study on the Asian A. cerana and the European A. mellifera. While the overall results and conclusions seem sound, there are a number of concerns to improve the reading and facilitate the interpretation of the results. The reference list should be revised to provide DOI indications. Some minor revisions are also listed.

Major comments:

Regarding the Materials and Method section.

• Page 5, line 106: the authors are mentioning that honeybee larval collection was performed in a single hive. Why the authors did not collect larvae from different hives? Is there any explanation for selecting only one hive to perform all their experiments? Even if it is a lot of work often on such experiments it is recommended to have more than one hive as source of bees.

• Page 6, line 119: Why the authors are using an anticoagulant buffer? How they managed the presence of fat body material?

• When working on the hemocytes, the authors used two different procedure of centrifugation. line 2126 page 6 it is at 1,000g for 1 minute while line 136 at page 7, the speed is reported to be 800g for 5 min. Why such a difference?

• Lines 140 and 144 page 7: the authors are washing cells under different conditions 3 or 2 times in PBS. Why such a difference?

• Did the authors evaluated the nodules attached to the larval tegument? They only mentioned trachea and fat body? Do they have evidences that there is no nodules attached to this structure.

• Page 8, line 172: Total RNAs were extracted from the larvae using Trizol. What about the RNAs from pupae and adults? As the authors used L5 instar larvae, 2-3 days pupae and young worker bees (line 170).

•

Result section

• I would suggest to change the order of the figures 1B and 1C, dose mortality curves first and then the times-mortality curves. A suggestion to help the reader, the author may add a sign to localize the LT50 and the LC50 values by broken lines

• Subsection 3.2 “ Comparative analysis of immune responses of two honeybees. The authors used two types of bacterial preparations (live bacteria and heat killed bacteria which is an important point. Why did the authors not included an additional control with a non-bacterial materials as it is often the case to follow nodulation?

• The authors assessed the expression level of genes (ApoLpIII, Def) encoding the two antimicrobial peptides (AMPs) Apolipophorin III and Defensin. Even if P. larvae is a Gram positive bacteria it would have been interesting to follow also apidaecin, abaecin and hymenoptaecin genes. The three last genes are reference genes to follow the impact of stressors on honeybee immunity.

• Is the injection of DEX to larvae has an incidence on the survival of the injected bees? The authors are mentioning that eicosanoids mediate both cellular and humoral immune responses in honeybees. Can the authors mentioned what are the results supporting the mediation of the cellular responses and those supporting the effect of eicosanoids on the humoral response?

The discussion section is hard to follow and should be revized by reducing the long discussion on PLA2 from other species. The message should be kept on honeybees and the interesting results the authors are bringing to the field of honeybee immunity

Minor revisions needed before it can be published.

It is recommended for each section when a species name is mentioned to write the full name, for example Apis mellifera and not A. mellifera etc in the M&M section

Can the authors confirmed that only Yonggyun Kim (YK, page 17, line 388) did the conceptualization of this work?

Figure 2 caption: Definition of PI used in figure 2 (D) red bars, THC definition, include values of percentage in fig. 2C.

Figure 3: ApoLpIII and Def should be in italic as they are working on gene expression.

Figure 5, the purple color should be change to be able to read the letter included, the size of the definition of the different letters can be optimized ofr a better reading, there is space for that.

Figure 7 is confusing as three subfigures are proposed (A, B and C) while only two are mentioned in the text (page 12, lines 267-271). Figure 7C according to the figure legend I have “lavel” should be changed for “level” when needed.

Reviewer #2: The current study presents some significant findings on the immune responses of two honey bee species to P. larvae infection. Here, I have raised up some concerns listed below:

L21. Should use the singular form of mortality.

L24-25. The sentence is confusing. Would be correct if it is written as “Hemocytes…..behaviors, accompanied by cytoskeletal extension and F-actin growth”?

L25. Redundant expression by saying “upon P. larvae infection” again.

L29. Use the italic format of “PLA2”.

L35. Use the italic format of “PLA2s”.

L54. Remove the definite article “the” before “American foulbrood”.

L93. Better use “in two honeybee species”.

L95. Use past tense for “propose”.

L126, 134 and 152. You stated that L5 instar larvae were used for hemocyte counts, hemocyte-spreading assay, and nodulation assy. Since L5 instar of honey bee larvae lasts over four and a half days, and can be divided into 9 periods including 3 feeding phases (LF1, LF2 and LF3), 3 cocoon-spinning phases (LS1, LS2 and LS3), and 3 pre-pupal phases (PP1, PP2 and PP3) (Michelette and Soares, 1993, Apidologie), which specific stage(s) did you harvest larvae from? Larvae from different L5 periods may differ in physiology and responses to P. larvae infection.

L156. Why did you incubate the larvae at 25℃? It is not the most suitable temperature for larval development.

L170. It is not common to describe the pupal development using the absolute days, but instead, using the morphological features like white-eyed, pink-eyed, is more popular and feasible.

L187. Independent samples?

L191. What age/stage were the test larvae?

L195. Should say “ten larvae were used in each treatment.”

L202-203. What was going on after the dsRNA delivery? The full information should be provided. Specifically, what’s the control and how was it performed?

L207. Generally, three biological replicates were very limited numbers for statistical analyses.

L214-215. Should use passive voice.

L217-220. Please double-check the LT50 and LC50 data in the text, which are not consistent with the data showed in Fig 1. Additionally, is there any significant differences in susceptibility to P. larvae between Am and Ac?

L222. Should be two honey bee species.

L263-265. The expression is very confusing. How to understand the regulation of PLA2B?

L278. Are the controls different between fig 8A and fig 8B?

L282. Should be “treatments”.

L322. Should be two honey bee species.

L329. Use the italic format of “Vg”.

For the entire Results section, the details of statistical outputs are missing, which should be provided.

6. PLOS authors have the option to publish the peer review history of their article (what does this mean?). If published, this will include your full peer review and any attached files.

Reviewer #1: No

Reviewer #2: No

---

## [Author Response · Author response to Decision Letter 0]

20 Nov 2023

[Reviewer #1]

Comment #1-1: Page 5, line 106: the authors are mentioning that honeybee larval collection was performed in a single hive. Why the authors did not collect larvae from different hives? Is there any explanation for selecting only one hive to perform all their experiments? Even if it is a lot of work often on such experiments it is recommended to have more than one hive as source of bees.

Response: All experimental larvae were collected from a hive to assess immunological assessment. This allowed us to minimize any genetic difference among individual larvae derived from different hives.

Comment #1-2: Page 6, line 119: Why the authors are using an anticoagulant buffer? How they managed the presence of fat body material?

Response: To minimize melanization of hemolymph, ACB was used. Under microscopic examination of the collected hemolymph, we did not detect any fat body cells. We add this information to the M&M.

Comment #1-3: When working on the hemocytes, the authors used two different procedure of centrifugation. line 2126 page 6 it is at 1,000g for 1 minute while line 136 at page 7, the speed is reported to be 800g for 5 min. Why such a difference?

Response: Both are corrected as follows: “1,000 x g for 3 min”

Comment #1-4: Lines 140 and 144 page 7: the authors are washing cells under different conditions 3 or 2 times in PBS. Why such a difference?

Response: Both are different washing steps. The first was to clean-up the remaining Triton-X100 while the latter was to replace the solution for mounting. Thus, the different washing frequencies were applied.

Comment #1-5: Did the authors evaluated the nodules attached to the larval tegument? They only mentioned trachea and fat body? Do they have evidences that there is no nodules attached to this structure.

Response: The pictures (Fig. 3A) show the nodules on trachea and fat body. However, we counted nodules in entire tissues of the test larvae.

Comment #1-6: Page 8, line 172: Total RNAs were extracted from the larvae using Trizol. What about the RNAs from pupae and adults? As the authors used L5 instar larvae, 2-3 days pupae and young worker bees (line 170).

Response: Rephrased as follows: “L5 instar larvae, 2-3 day old pupae, and young worker bees less than 1 week old after emergence were used for total RNA extraction after removing their intestines to avoid any contamination derived from non-target organisms using Trizol reagent (Invitrogen, Carlsbad, CA, USA) according to the manufacturer’s instructions.”

Comment #1-7: I would suggest to change the order of the figures 1B and 1C, dose mortality curves first and then the times-mortality curves. A suggestion to help the reader, the author may add a sign to localize the LT50 and the LC50 values by broken lines

Response: To measure LC50 at the specific time after treatment, LT50 was first measured in Fig. 1B. To make clear this point, we add the treated bacterial concentration and the time to measure the virulence. 

Comment #1-8: Subsection 3.2 “ Comparative analysis of immune responses of two honeybees. The authors used two types of bacterial preparations (live bacteria and heat killed bacteria which is an important point. Why did the authors not included an additional control with a non-bacterial materials as it is often the case to follow nodulation?

Response: We used non-bacterial treatment, which was denoted as ‘CON’. We mentioned this in the figure caption: Control (‘CON’) used sterilized PBS for injection.

Comment #1-9: The authors assessed the expression level of genes (ApoLpIII, Def) encoding the two antimicrobial peptides (AMPs) Apolipophorin III and Defensin. Even if P. larvae is a Gram positive bacteria it would have been interesting to follow also apidaecin, abaecin and hymenoptaecin genes. The three last genes are reference genes to follow the impact of stressors on honeybee immunity.

Response: This is a nice suggestion. However, our purpose in this study was to demonstrate any humoral immune response in this honeybees. As you see in Fig 3C, we showed the immune response. 

Comment #1-10: Is the injection of DEX to larvae has an incidence on the survival of the injected bees? The authors are mentioning that eicosanoids mediate both cellular and humoral immune responses in honeybees. Can the authors mentioned what are the results supporting the mediation of the cellular responses and those supporting the effect of eicosanoids on the humoral response?

Response: The down-regulation of both immune responses by the DEX treatment was rescued by the addition of arachidonic acid (AA). AA is the precursor of the eicosanoid biosynthesis.

Comment #1-11: The discussion section is hard to follow and should be revized by reducing the long discussion on PLA2 from other species. The message should be kept on honeybees and the interesting results the authors are bringing to the field of honeybee immunity

Response: Our discussion consists of two parts: one (lines 286-330) for immune responses of the honeybees and the other (lines 331-386) for PLA2-based eicosanoid role in the immune responses. The comparative analysis of honey bees and other insects in PLA2 genes are necessary to validate the honeybee PLA2s.

Comment #1-12: It is recommended for each section when a species name is mentioned to write the full name, for example Apis mellifera and not A. mellifera etc in the M&M section

Response: I am not sure how to write the scientific names in PLoS. When we read the previous papers published in PLoS, the first scientific names are fully-written and the next citation used acronyms in genus names.

Comment #1-13: Can the authors confirmed that only Yonggyun Kim (YK, page 17, line 388) did the conceptualization of this work?

Response: This study was devised by Y Kim. In this regard, he alone was designated in the category. 

Comment #1-14: Figure 2 caption: Definition of PI used in figure 2 (D) red bars, THC definition, include values of percentage in fig. 2C.

Response: All the parameters are explained in the figure caption as follows:

“….indicates 10 μm. (B) Comparison of the total hemocyte count (THC) between two species. ‘NS’ stands for no significance. (C) Differential hemocyte counts. The statistical analysis was performed by �2 test to compare the hemocyte composition between two species. (D) Hemocyte-spreading behavior. Each larva was injected with 1 μL of P. larvae (‘Pl’, 5 � 104 cells). Each measurement for the spreading behavior used 100 randomly chosen hemocytes.”

Comment #1-15: Figure 3: ApoLpIII and Def should be in italic as they are working on gene expression.

Response: Corrected as suggested

Comment #1-16: Figure 5, the purple color should be change to be able to read the letter included, the size of the definition of the different letters can be optimized ofr a better reading, there is space for that.

Response: Corrected as suggested

Comment #1-17: Figure 7 is confusing as three subfigures are proposed (A, B and C) while only two are mentioned in the text (page 12, lines 267-271). Figure 7C according to the figure legend I have “lavel” should be changed for “level” when needed.

Response: Corrected as suggested

 

[Reviewer #2]

Comment #2-1: L21. Should use the singular form of mortality. 

Response: Corrected as suggested

Comment #2-2: L24-25. The sentence is confusing. Would be correct if it is written as “Hemocytes…..behaviors, accompanied by cytoskeletal extension and F-actin growth”?

Response: Corrected as suggested

Comment #2-3: L25. Redundant expression by saying “upon P. larvae infection” again.

Response: Deleted

Comment #2-4: L29. Use the italic format of “PLA2”.

Response: Four individual genes names followed the italic format.

Comment #2-5: L35. Use the italic format of “PLA2s”.

Response: Four individual genes names followed the italic format.

Comment #2-6: L54. Remove the definite article “the” before “American foulbrood”.

Response: Deleted

Comment #2-7: L93. Better use “in two honeybee species”.

Response: Corrected as suggested

Comment #2-8: L95. Use past tense for “propose”.

Response: Corrected as suggested

Comment #2-9: L126, 134 and 152. You stated that L5 instar larvae were used for hemocyte counts, hemocyte-spreading assay, and nodulation assy. Since L5 instar of honey bee larvae lasts over four and a half days, and can be divided into 9 periods including 3 feeding phases (LF1, LF2 and LF3), 3 cocoon-spinning phases (LS1, LS2 and LS3), and 3 pre-pupal phases (PP1, PP2 and PP3) (Michelette and Soares, 1993, Apidologie), which specific stage(s) did you harvest larvae from? Larvae from different L5 periods may differ in physiology and responses to P. larvae infection.

Response: To clarify this issue, we add following sentence in the M&M: “Larval stage used 2 days-old L5 individuals.”

Comment #2-10: L156. Why did you incubate the larvae at 25℃? It is not the most suitable temperature for larval development.

Response: To reduce the variation in the nodule formation depending on varying ambient temperature, we used a constant temperature. This explanation is added to the M&M.

Comment #2-11: L170. It is not common to describe the pupal development using the absolute days, but instead, using the morphological features like white-eyed, pink-eyed, is more popular and feasible.

Response: We measured the pupal age in days.

Comment #2-12: L187. Independent samples?

Response: Corrected as suggested

Comment #2-13: L191. What age/stage were the test larvae?

Response: L3 larvae were used. This is added to the M&M.

Comment #2-14: L195. Should say “ten larvae were used in each treatment.”

Response: Corrected as suggested

Comment #2-15: L202-203. What was going on after the dsRNA delivery? The full information should be provided. Specifically, what’s the control and how was it performed?

Response: Corrected as suggested

Comment #2-16: L207. Generally, three biological replicates were very limited numbers for statistical analyses.

Response: The limited replication number indicated the significant treatment effect in this assay.

Comment #2-17: L214-215. Should use passive voice.

Response: Corrected as suggested

Comment #2-18: L217-220. Please double-check the LT50 and LC50 data in the text, which are not consistent with the data showed in Fig 1. Additionally, is there any significant differences in susceptibility to P. larvae between Am and Ac?

Response: Corrected by exchanging LC50 values between two species. The median values were not statistically different. These are included in the text.

Comment #2-19: L222. Should be two honey bee species.

Response: Corrected as suggested

Comment #2-20: L263-265. The expression is very confusing. How to understand the regulation of PLA2B?

Response: As mentioned in discussion, PLA2B is classified into Group XII as a sPLA2. Its physiological function may be not venom unlike PLA2A. It may be involved in adult reproduction. This is mentioned in the discussion.

Comment #2-21: L278. Are the controls different between fig 8A and fig 8B?

Response: In Fig 8A, control is changed into dsCON

Comment #2-22: L282. Should be “treatments”.

Response: Corrected as suggested

Comment #2-23: L322. Should be two honey bee species.

Response: Corrected as suggested

Comment #2-24: L329. Use the italic format of “Vg”.

Response: Corrected as suggested

---

## [Decision Letter · Decision Letter 1]

2 Jan 2024

Phospholipase A2 activity is required for immune defense of European (Apis mellifera) and Asian (Apis cerana) honeybees against American foulbrood pathogen, Paenibacillus larvae

PONE-D-23-26300R1

Dear Dr. Kim,

We’re pleased to inform you that your manuscript has been judged scientifically suitable for publication and will be formally accepted for publication once it meets all outstanding technical requirements.

Kind regards,

Olav Rueppell

Academic Editor

PLOS ONE

Additional Editor Comments (optional):

Reviewers' comments:

Reviewer's Responses to Questions

**Comments to the Author**

1. If the authors have adequately addressed your comments raised in a previous round of review and you feel that this manuscript is now acceptable for publication, you may indicate that here to bypass the “Comments to the Author” section, enter your conflict of interest statement in the “Confidential to Editor” section, and submit your "Accept" recommendation.

Reviewer #1: All comments have been addressed

Reviewer #2: (No Response)

2. Is the manuscript technically sound, and do the data support the conclusions?

Reviewer #1: Yes

Reviewer #2: Yes

3. Has the statistical analysis been performed appropriately and rigorously? 

Reviewer #1: Yes

Reviewer #2: Yes

4. Have the authors made all data underlying the findings in their manuscript fully available?

Reviewer #1: Yes

Reviewer #2: Yes

5. Is the manuscript presented in an intelligible fashion and written in standard English?

Reviewer #1: Yes

Reviewer #2: Yes

6. Review Comments to the Author

Reviewer #1: The different responses of the authors to the reviewer #1 comments and suggestions are almost acceptable.

The answer to comment #1.1 is partially satisfactory in terms of genetic difference but not really in terms of biological replicates coming from different hives.

Comment # 1.2: Melanizatioin process is not linked to coagulation. There is other chemicals that are helpful to minimise melanization, e.g. PMSF (even if its half life is rather short).

Comment #1.4, three washing steps would have been better whatever the components to remove (Triton X100 or ACB).

For the other comments and suggestions, the responses of the authors are satisfactory.

Reviewer #2: (No Response)

7. PLOS authors have the option to publish the peer review history of their article (what does this mean?). If published, this will include your full peer review and any attached files.

Reviewer #1: No

Reviewer #2: No

---

## [Editor Report · Acceptance letter]

29 Jan 2024

PONE-D-23-26300R1 

PLOS ONE

Dear Dr. Kim, 

I'm pleased to inform you that your manuscript has been deemed suitable for publication in PLOS ONE. Congratulations! Your manuscript is now being handed over to our production team.

Kind regards, 

on behalf of

Dr. Olav Rueppell 

Academic Editor

PLOS ONE